# Impacts of Video Display on Purchase Intention for Digital and Home Appliance Products—Empirical Study from China

**Ruohong Hao [1], Bingjia Shao [2,3] and Rong Ma [4],***

[1]  School of Economics and Management, Tongji University, Shanghai 200092, China; 1930424@tongji.edu.cn
[2]  School of Economics and Business Administration, Chongqing University, Chongqing 400044, China; shaobingjia@cqu.edu.cn
[3]  Chongqing Key Laboratory of Logistics at Chongqing University, Chongqing 400044, China
[4]  School of Economics and Management, Yunnan Agricultural University, Kunming 650201, China
*  Correspondence: 2011019@ynau.edu.cn; Tel.: +86-187-2517-8125

**Abstract:** Rapid online trading expansion and the bloom of internet technologies has raised the importance of effective product video presentations for online retailers. This article developed a model for the impacts of video presentations on purchase intention for digital and home appliance products. Four group experiments were designed, and empirical tests were performed. This research found that presenting videos on how to use digital and home appliance products increased purchase intention by raising the information gained by customers. Meanwhile, video tutorial information had insignificant effects related to the knowledge and experience of customers.

**Keywords:** video presentations; e-commerce; purchase intention

---

## 1. Introduction

Videos have gradually come to dominate online product displays. According to statistics from JD.com, the number of product videos increased by 145 times in 2018, covering 80% of active shops on JD.com. Furthermore, a 2019 consumer survey showed that consumers would like to view more product pictures and videos when shopping online. This means that video displays are rapidly developing, and consumers are getting used to watching product videos to get more information about product appearance, features, and so on. Consumers need to physically inspect or try on the product to make more informed judgements. When shopping online, the level of consumers' perceived risk may be increased because they cannot touch the physical products [1,2]. Although many consumers have already tried to purchase online, they will still hesitate to pay the next time [3]. Therefore, product information plays an important role in consumers' purchasing decisions [4]. According to media theories, compared to text and still images, videos, which can provide multiple cues and richer information, will allow recipients to understand the product better [5]. Video displays, which give consumers a more realistic experience by providing more specific information, increase purchase intentions by reducing consumers' perception of risk [6].

Different types and different presentation formats will have different effects on consumer willingness to purchase online. Part of the existing research has focused on comparing the impacts of different online product display forms on consumers' perceptions or behaviors. Aljukhadar and Senecal compared the impacts of text and video displays on the trust, arousal, and quality of information as perceived by the consumers [7]. Some studies compared static pictures, videos, and other display forms [8–11]. However, these studies focused on the impacts of display form (e.g., through studying the content of online product videos) and rarely combined display form

and display content. Other studies have focused on the impact of online product presentations on products with different attributes. Researchers usually divide products into experience and search products [12–14] or hedonic and utilitarian products [15]. For example, Huangjing classified products as experience and search products and explored the impacts of different display forms on consumer product reviews [14]. Roggeveen classified products as hedonic and functional products and explored the impacts of different display forms on consumers' preferences [15]. Furthermore, some studies have focused on the specific product type, especially the impact of online product presentations on clothing purchases [1,4,16]. However, studies on the impact of online product presentations on other product types has not been explored. Therefore, this paper focused on home digits and appliance products and used the stimulus–organism–response (S–O–R) model to explain the impacts of different video contents on consumers' perceptions and purchase intentions. The main contributions of this paper include (1) exploring how usage scenario information and usage tutorial information in online product presentation videos affect consumers' perceived information volume. (2) exploring how consumers' perceived information volume affect their purchase intention. We investigated how video displays affect consumers' purchase intentions. It was found that the use of videos for home devices and appliances influenced the level of information gained by the consumer: the more the volume of perceived information, the more the perceived value. Purchase intention will increase when consumers believe that the benefits they can get from purchasing the product outweigh the price of the product. On the one hand, this research provides information on the mechanisms by which video displays influence consumer behaviors; on the other hand, these results will help retailers to design product videos to promote sales.

## 2. Literature Review

### 2.1. Online Product Display

Online product displays are a basic IT tool used by retailers [17]. From the perspective of communication, online product displays are defined as a special type of communication that can be used by online retailers to communicate with customers and provide product information [18]. In related research on online product displays, scholars focused on comparing the differences between display forms and found that video displays, compared to text, had a significant impact on users that browse for entertainment [8]. Some experts believe that visual displays, such as videos, can affect consumer emotions more strongly compared to text, which is beneficial for the emotional aspect of the consumer relationship experience [19]. Some scholars have compared the influences of static pictures, dynamic video, and other display forms (such as slides) on consumers' preferences. Roggeveen et al. compared the three product display forms—static pictures, slides, and dynamic video—and found that dynamic display forms were more likely to make consumers to choose hedonic products [15]. However, in the field of social media, some people think that the influences of pictures and videos on consumer engagement can only produce compliant participation rather than interactive participation, but the change of consumer cognition brought by text links will make consumers participate more actively [20]. Nonetheless, few existing studies have explored the impact of video display on consumer behavior by analyzing video content. In fact, video displays contain rich color, visual cues, dynamic movement, and various sounds, transmitting rich information to catch consumers' attention [21].

### 2.2. Impacts of Videos on Different Products

Different online display forms lead consumers to make different purchase decisions for different products [22]. In existing research, scholars classified products as search products and experience products depending on product attributes. Some scholars believe that consumers rate search products under a static display more highly, while consumers rate experience products under a dynamic display more highly [15]. Some scholars have classified products as functional and hedonic and believe that the dynamic display form (video) enhances consumers' preference for experiential products [19]. However,

some scholars claim that consumers need different information for different categories of products. Taking digital appliances as an example, consumers not only require product parameters with more details but also take the appearance of products into consideration, so they require retailers to provide comprehensive and objective information [23]. Using the relevant data from China's consumer market, we found that the national consumption level is continuing to increase, and consumers' demand for online purchases of digital home appliances is growing. In 2018, the retail sales of B2C home appliances (including mobile terminals) in China increased by 17.5% over the previous year, including a 19.42% growth in mobile terminal devices such as mobile phones and tablets. People are increasingly preferring to purchase digital home appliances online. Therefore, this paper focuses on digital home appliances and explore the impact of video display form on such products.

### 2.3. Impacts of Video Display on Consumer Perceptions

Online product display is used to introduce products to consumers and help them to have a clear understanding of products [10]. In the online environment, it is impossible for consumers to touch or check physical products, so a good online product image can help consumers to identify and understand products [5]. Therefore, online product display, which can influence consumers' perceptions and help them to make relevant purchase decisions, is very important for retailers. The vividness theory [24] states that vivid information will produce more images of products in people's minds and increase their imaginal consumption [25]. The dynamic display form is a more vivid display form. Compared to the static display form, it makes consumers feel more like participants and enables them to imagine the feeling of using the products better [19]. Therefore, we believe that video display, as a form of dynamic display, can influence consumers' cognition of products and ultimately influence consumers' purchase intentions or behaviors.

### 2.4. S–O–R Model

The S–O–R (stimulus–organism–response) model is used to explore the influences of the physical environment on consumer behavior. Eroglu et al. modified the model under the context of online retail, where the "stimulus" is a synthesis of all the visible and audible cues for online shoppers. The organism is the internal emotional and perception status. "Reaction" refers to the emotions and perceptions generated in the process of Internet browsing that make consumers react [26]. The S–O–R model is widely used to explore the impacts of online display on consumers. Jeong used the S–O–R model to explore the impacts of online product display on consumer website patronage intentions. Online product displays stimulate consumers on four levels (entertainment, educational, escapist, and esthetic) and also influence consumers' (arousal and pleasure) emotions, eventually influencing the website patronage intentions of consumers [27]. Fiore et al. established the S–O–R model to explore how the level of interactive technology influences consumers' attitudes to retailers, online purchase intentions, and the willingness to shop with the same retailer again [28]. Based on the S–O–R model, this paper explores the impact of video display on consumers' purchase intentions and the impact of video content on consumers' perceptions. Finally, it investigates how video display influences consumers' purchase intentions. Table 1 shows the content of the literature review.

**Table 1.** Literature review table.

| | Author | Year | Content |
|---|---|---|---|
| Online Product display | Li, Wei, Tayi, et al. [17] | 2016 | Online product display is a basic IT tool, used by retailers. |
| | Wang, Cui, Huang, et al. [18] | 2016 | From the perspective of communication, online product display is defined as a special communication, which can be used by online retailers to communicate with customers and provide product information |
| | Pera, Viglia [19] | 2016 | Visual display such as video can affect consumers' emotions compared to text, which is beneficial to the emotional level of consumer relationship experience |
| | Viglia, Pera, Bigné [20] | 2017 | In the field of social media, some people think that the influence of pictures and videos on consumer engagement can only produce compliant participation rather than interactive participation, but the change of consumer cognition brought by text link will make consumers participate more actively. |
| | Xu, Chen, Santhanam [5] | 2015 | Video displays contain rich color, visual cues, dynamic movement and various sounds, transmitting rich information to catch consumers' attention. |
| Impacts of Videos on Different Products | Roggeveen, Grewal, Townsend, et al. [15] | 2015 | This paper classified products as hedonic products and functional products and explored impacts of different display form on consumers' preference. |
| | Huang, Zou, Liu, et al. [14] | 2017 | This paper classified products as experience products and search products, and explored impacts of different display form on consumers' product review |
| | Zhang, Chen [21] | 2006 | Different online display forms make consumers having different purchase decisions for different products when making purchase decisions. |
| | Tang [22] | 2012 | Taking digital appliances as an example, consumers will not only require product parameters with more details, but also take the appearance of products into consideration, so they will require retailers to provide comprehensive and objective information |
| Impacts of Online product presentation video on Consumer Perceptions | Suh, Kim, Kim [7] | 2018 | The complexity of product materials and perceived knowledge will affect consumers' choice of produce presentation forms. For electronic products, consumers prefer to obtain the information of product hardness and quality by watching videos; For the diversity and problems of clothing materials, consumers prefer to view by zooming. |
| | Aljukhadar, Senecal [8] | 2017 | Streaming video method can stimulate consumers to reach a higher level of arousal, trust competence, and information quality than text when consumers browsing with a recreational goal. |
| | Overmars, Poels [9] | 2015 | By comparing four different presentation formats (static interface, interactive interface, video interface, and actual product), it is confirmed that consumers can perceive the tactile sensation in real environment by simulating touch gestures with interactive interface. |
| | Jiang, Benbasat [10] | 2007 | By comparing multiple still pictures, video with narration, video with narration and virtual product experiences, it is conducted that vividness and interaction will have an impact on consumers' cognition and emotion. |
| | Yue, Liu, Wei [11] | 2017 | This paper defined website presents product with picture, video and 3D image means as high media richness and confirmed that the influence of media richness on consumers' perceived risk and trust. And |

**Table 1.** *Cont.*

|  | Author | Year | Content |
|---|---|---|---|
|  | Flavian, Gurrea, Orus, et al. [23] | 2015 | In the online environment, it is impossible for consumers to touch or check physical products, so a good online product image can help consumers to identify and understand products. |
|  | Nowlis, Mandel, Mccabe [25] | 2016 | Vivid information will produce more images of products in people's minds and increase the imaginal consumption |
| S–O–R Model | Eroglu, Machleit, Davis [26] | 2001 | This paper modified the model under the context of online retail, where the "stimulus" is a synthesis of all the visible and audible cues for online shoppers. |
|  | Jeong, Fiore, Niehm, et al. [27] | 2009 | This paper explored impacts of online product display on consumer website patronage intention, online product display stimulate consumers on four levels (entertainment, educational, escapist, and esthetic) and also influence on consumers' (arousal and pleasure) emotion, eventually influencing the website consumer website patronage intention. |
|  | Fiore, Kim, Lee [28] | 2010 | This paper established the S–O–R model to explore how the level of image interactive technology influences the attitude to retailer, online purchase intention and the willingness to shop with the same retailer again. |

## 3. Theoretical Framework

The content of various short videos, collected across multiple digital and home appliance online retailer platforms was analyzed to explore the display features of product videos.

According to "The 2018 Analysis Report of Home Appliance Online Purchase" statistics, the websites with the TOP 3 market share from the home appliance online sales market are JD.com, with a share of 60%; Tmall.com, accounting for 28%; and Suning.com, with a 1% share. According to the "2017 Digital Products Consumption Tendency Report", the following 10 categories were picked for sampling: mobile phones, televisions, tablets, laptops, rice cookers, air conditioning systems, smart bands, earphones, washing machines, and electric shavers. Most people purchased items from one or more of these categories. These 10 categories are more common in daily life, and their marketing is developed and can reflect the general principles of short video display for digital and home appliance products.

One hundred short videos were picked from the Top 3 online retailers. The sampling principles used were (1) based on market share—the videos for each category were picked from the Top 3 retailers in the proportion 6:3:1; and (2) products were searched by keywords separately on each retailer's platform, and the search results were sorted by sales volume. Products with short videos in the search result were picked as samples. All searches used a newly registered account to avoid purchasing history influences.

The content of these 100 collected short videos included product appearance, features, usage scenarios, video tutorial information, the production process, brands, and hosts. Products' appearance display had the largest proportion; nearly 90% of collected videos showed the appearance of products. Next was brand information: 81 out of 100 videos represented brand information. Feature display was third with 56%. Fourth was the usage scenario display, accounting for 37 out of 100 videos. After this, 25% of videos were delivered with a video tutorial information and 13% of videos showed the production process. Only three videos contained information about the host. The statistics are shown in Figure 1.

The above investigation found that video displays for digital and home appliances not only presented the products' appearances, features, and brands but also the products' usage scenarios, tutorial information, production process, and host. The usage scenarios demonstrate the proper situations for using products, which outline the products' features. Video tutorial information gives

consumers guidance for using products. A small part of the sample of short videos contained products' process and host information. This might mean that online consumers are more likely to get more information about products, including what benefits the products will give. This article selects usage scenarios and tutorial information as two separate key factors to explore its impacts on consumers' purchasing intentions.

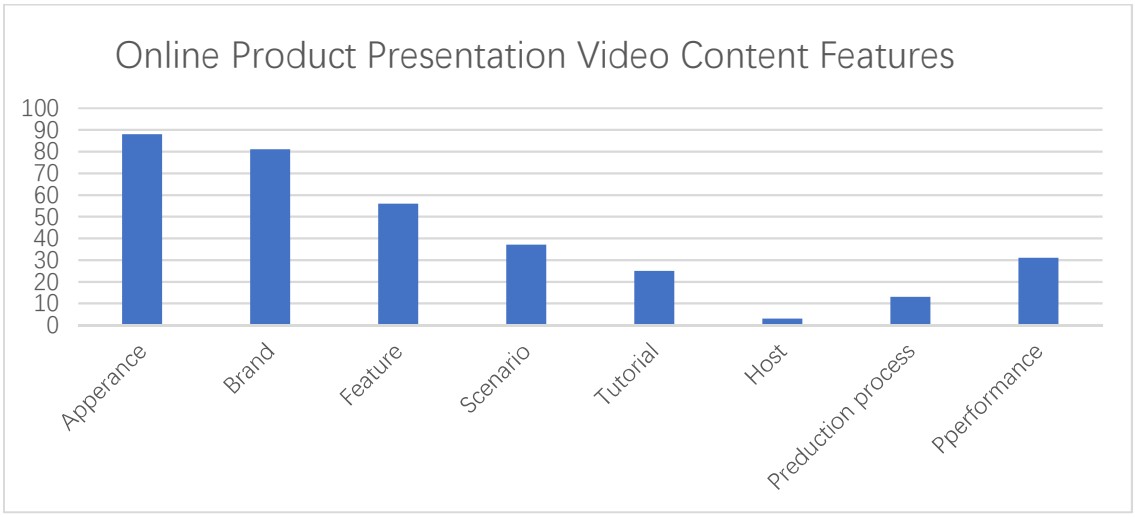

**Figure 1.** The classified statistics of short video content features.

### 3.1. Usage Scenarios and Perceived Information Volume

The scene generally refers to the spatial environment, while the situation generally refers to the behavioral scene or psychological atmosphere, and the generalized scene describes the situation [29]. Product usage scenarios refer to the spatial environment and behavioral scene. Many videos for digital and home appliance products display products in a specific scenario to deliver more product information to consumers. The perceived information volume is the positive information that the consumer obtains through watching short videos. In marketing activities, retailers often provide positive information to consumers to lift sales. Normally, consumers prefer to search for internal and external information before making purchase decisions. Consumers are stimulated by external information and then try to search internal information from their memory. The correlation with external and internal information will ultimately affect consumers' willingness to purchase [30].

The situational information shown in videos combines video animation and audio methods to convey more information that cannot be conveyed by static display. Placing products in a specific usage scenario will stimulate consumers, lead to the development of product associations, and send more product information to consumers, who are exposed to a greater volume of information. Thus, the following hypothesis is made:

**Hypothesis 1 (H1).** *The display of product usage scenario information will positively affect the information perceived by consumers.*

### 3.2. Tutorial Information and Perceived Information Volume

A tutorial is an introduction for how to use a product. As the majority of digital and home appliances are high-tech products, consumers need to pay certain learning costs. Therefore, displaying tutorial information in a video helps consumers to gain knowledge of products quickly and stimulates the consumer to search for internal information, leading them to increase their perceived information volume. Thus, the following hypothesis is made:

**Hypothesis 2 (H2).** *Presenting product usage tutorial information positively affects consumers' perceived information volume.*

*3.3. Perceived Information Volume and Purchase Intention*

Information plays an important role in TV shopping. The information presented on TV shopping networks, on the one hand, is obtained through internal searches and, on the other hand, is the only source of information consumers can get to make purchasing decisions. If consumers are given enough information, they can make purchasing decisions even if they cannot check the goods [31]. In the Internet context, the attributes of website advertisements, perceived information volume, and perceived entertainment have important impacts on consumers' purchasing decisions. If online shopping websites can provide a lot of information or cause a fun or emotional response to shopping, consumers can still make wise purchasing decisions [32]. In the online shopping scenario, compared to consumers' perceived risk, perceived information or emotions (pleasure or arousal) can better predict consumers' purchase intentions [33].

Video plays a role in information dissemination by conveying more information to consumers via visual and auditory methods to stimulate the senses. When consumers watch good videos, compared to static images or limited text, they receive more information, a variety of senses are stimulated, their association with the product increases, and their understanding of product information increases, thus generating a positive influence on consumers' purchase intentions. Thus, the following hypothesis is made:

**Hypothesis 3 (H3).** *Perceived information has positive impacts on purchasing intention.*

*3.4. Perceived Information, Perceived Risks, and Purchase Intention*

Perceived risk is a kind of uncertain feeling, where consumers believe that there will be some bad consequences after using a product or service [34]. Some research has found that consumers search for external clues to reduce uncertainty and perceived risk, as perceived information plays a significant role in consumers' decision making. Purchase intention increases when consumers perceive that there is less risk but more product information [33]. A lack of product information and experience lead to increases in uncertainty and perceived risk. If consumers can search for more information during the process of browsing products, perceived risk will decline, ultimately affecting consumers' purchasing behaviors [35].

The perceived information volume affects consumers' purchase intentions by influencing their perceived risks. When the perceived information volume increases, consumers can fully understand products, reducing their uncertainty and avoidance and, furthermore, reducing the perceived risk. When the perceived risk is at a low level, the possibility of consumers buying products will increase. Thus, the following hypothesis is made:

**Hypothesis 4 (H4).** *Perceived risk plays a mediating role between perceived information and purchase intention.*

*3.5. Perceived Information, Perceived Value, and Purchase Intention*

Compared to the product price, consumers' evaluation of the value of a product or service is its perceived value. Compared to consumer satisfaction, perceived value has a greater impact on a consumer's purchase decision [36]. In reality, if a consumer is fully satisfied with a product or service but the product price is too high, their purchase intention will decline. Perceived value increases only when consumer perceives that a product or service will give great benefits, influencing their purchasing intention positively [37]. The impacts of perceived information on perceived value have

been systematically studied, yet few empirical experiments have verified that perceived information has a significant positive influence on perceived value [38].

The more information perceived by consumers, the clearer the product value perceived by consumers when they shop online is. This finally increases their purchase intention and affects their purchase behavior. The perceived information volume affects the perceived value, influencing purchase intention. Thus, the following hypothesis is made:

**Hypothesis 5 (H5).** *Perceived value plays a mediating role between perceived information and purchase intention.*

Figure 2 summarizes the conceptual framework, showing the influence of an online product presentation video about digital and home appliance products on consumers' willingness to purchase, based on the S–O–R model. This framework proposes how online product presentation videos affect consumers' willingness to purchase by affecting their perceptions. Figure 3 shows the hypothesis framework of this article.

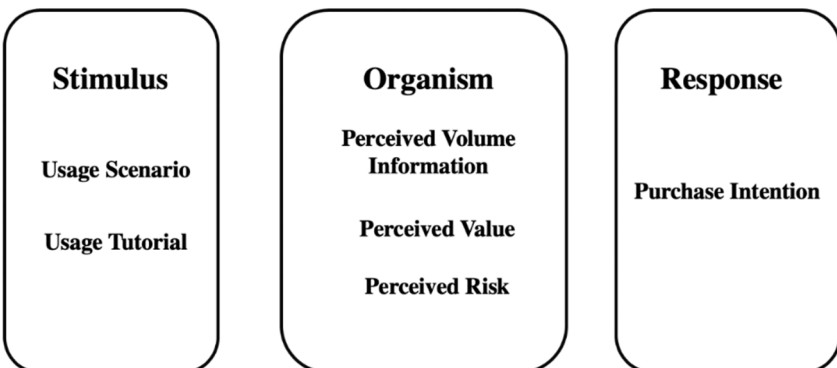

**Figure 2.** Conceptual framework.

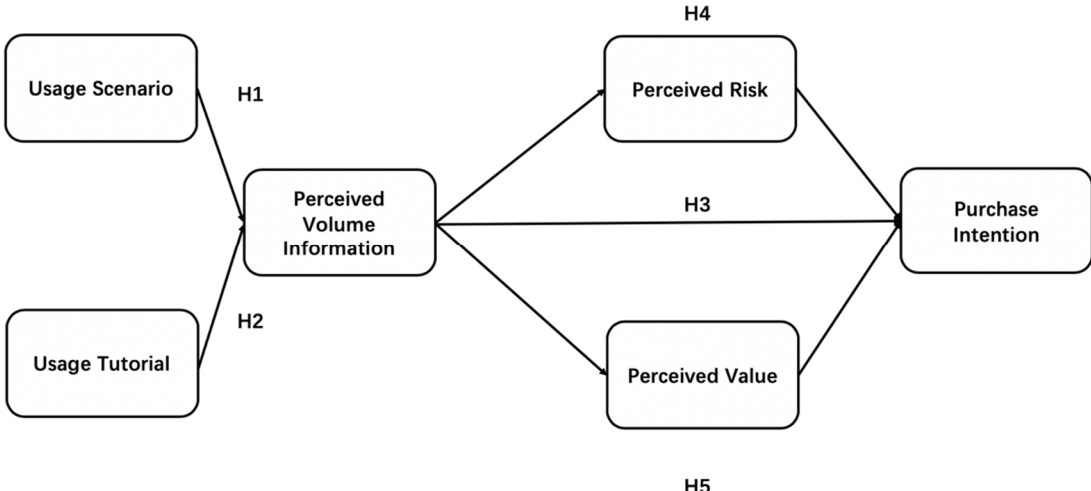

**Figure 3.** Hypothetical framework.

## 4. Methodology and Experiment Design

### 4.1. Experiment One: Product Usage Scenario

This experiment explored the influence of the perceived information volume on purchase intention when consumers watch short product videos. This was done by simulating a video display for digital home appliances under a real online shopping environment. The mobile phone category was selected

as the experimental sample. Mobile phones are popular products for all ages. The entertainment and search properties of mobile phones are important for their evaluation. Thus, imagination was the priority when determining the performance of products. Two experimental groups were designed: an experimental group without product usage scenario information and an experimental group with product usage scenario information. The aim was to alter the information perceived by consumers from the video by changing the product usage scenario information.

Experiment Design

Mobile phone A was selected as the sampled product. In order to ensure that other variables were consistent, the product information displayed by the experimental combination control group was consistent in terms of sound, and for the experimental group, the usage scene information of the product was displayed on the screen. Volunteers were told they were going to purchase a mobile phone and browse product information online. Volunteers completed a questionnaire based on their real feelings after watching short videos. Control group volunteers watched videos without product usage scenario information, as shown in Figure 4. The experimental group watched videos that included product usage scenario information, as shown in Figure 5.

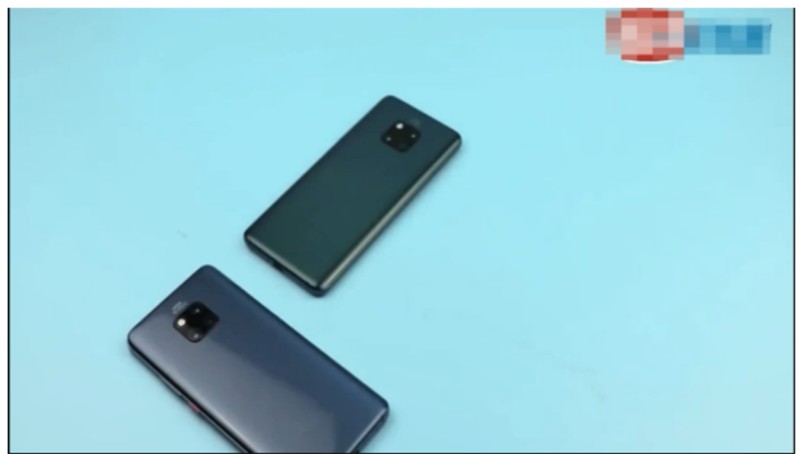

**Figure 4.** Example of information presented in the no usage scenario.

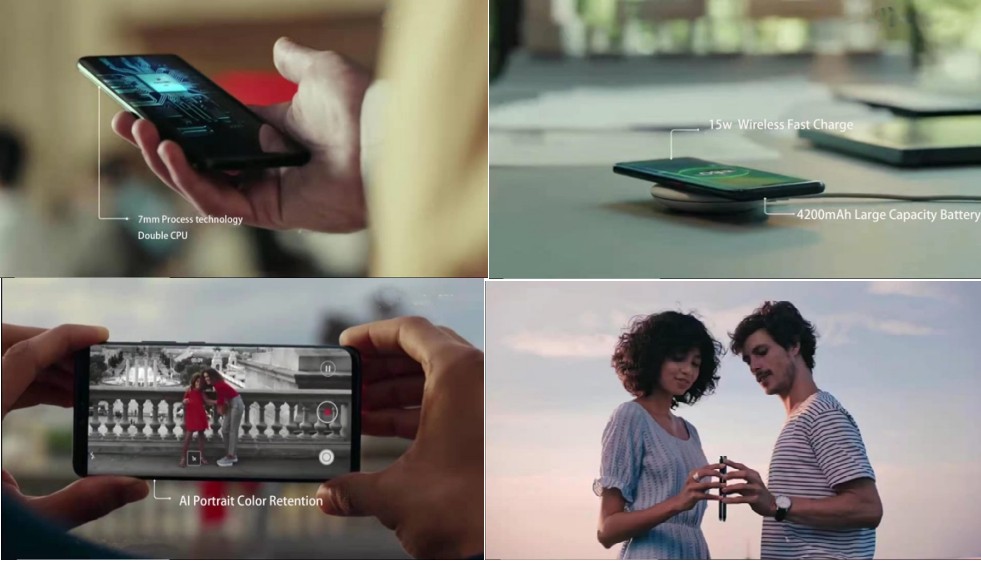

**Figure 5.** Example of usage scenario information.

The selected mobile phone A has the following features: fast charge, a nice portrait, and reverse charging. In order to effectively examine the impacts of product usage scenario information on consumers' perceived information volume, the videos that the control group watched only contained product appearance and product feature information. By contrast, the experimental group watched a video with product usage scenario information, for example, showing fast charging in the office, taking a portrait at a family gathering, and reverse charging other devices outdoors. This outlined the product's features.

### 4.2. Experiment Two: Product Tutorial Information

This experiment explored the impacts of video content on consumers' perceived information volume. A juice extractor was selected as the sampled product. This is a traditional home appliance that occupies a large part of the market; however, consumers are more likely to choose more fashionable appliances.

Experiment Design

One juice extractor was selected from the online sale platform. Volunteers were told that they were going to purchase a portable juice extractor and browse product information online. In order to ensure that other variables were consistent during the experiment, the video shown to the experimental group had no product usage tutorial information display, which was included in the control group's video. Volunteers completed a questionnaire based on their real feelings after watching short videos. The control group watched videos without product usage tutorial information, as shown in Figure 6, while the experimental group watched videos with product usage tutorial information, as shown in Figure 7.

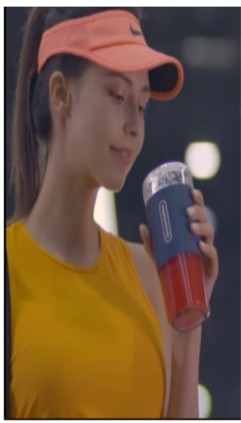

**Figure 6.** The example of no product tutorial information.

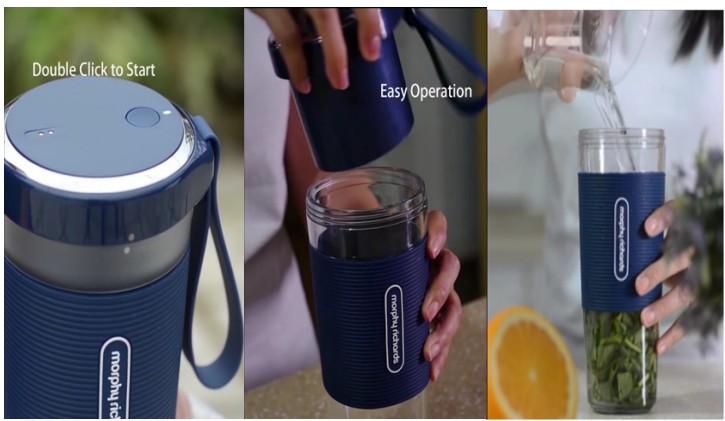

**Figure 7.** The example of using product tutorial information.

In this experiment, the tutorial information explained the entire juice extracting process. In order to effectively examine the impacts of product tutorial information on purchase intention, the videos that the control group watched only contained product appearance and product feature information. The experiment group watched videos with tutorial information added in.

### 4.3. Measurement Table Design

The measurement table used in this experiment was designed based on existing research. Detailed criteria are shown in Table 2.

**Table 2.** Measurements source.

| Factors | Code | Measures | References |
|---------|------|----------|------------|
| **Perceived information volume (PIV)** | PIV1 | Detailed product Description | Kim et al., 2010 [31–33] |
| | PIV2 | Get lots of product information | |
| | PIV3 | Be able to fully understand product information | |
| | PIV4 | Obtained information help to make purchasing decisions | |
| **Perceived risks (PR)** | PR1 | Concerning that the actual product does not match the displayed features | Kim et al., 2010 [31,32] |
| | PR2 | Concerning that actual product is not satisfactory | |
| | PR3 | Concerning that some of the functions of the actual products cannot achieve the advertised effect | |
| | PR4 | Concerning that using the product will affect how others perceive you | |
| | PR5 | Concerning about using the product and your friends will think you're funny | |
| | PR6 | Concerning product is inappropriate for using in public | |
| **Perceived value (PV)** | PV1 | make a good impression on others | Soutar, 2001 [39], William et al., 1991 [40] |
| | PV2 | recognized by other people | |
| | PV3 | Create a positive social image | |
| | PV4 | Like product | |
| | PV5 | Feel happy when using this product | |
| | PV6 | Enjoy using this product | |
| | PV7 | Want to own this product | |
| | PV8 | The actual price of this product is X, which is high cost performance | |
| | PV9 | Price is acceptable | |
| | PV10 | It's a good buy | |
| **Purchase Intention (PI)** | PI1 | Will choose to by this product | Kim et al., 2010 [33], William et al.,1991 [40] |
| | PI2 | glade recommend this product to people | |
| | PI3 | If this product is needed, happy to buy it | |
| | PI4 | Choose the same when next purchase | |

A questionnaire was designed based on the above measurement table and modified after the pre-experiment. A seven-level Likert scale was used in this experiment.

*4.4. Data Collection*

4.4.1. Experiment One

The questionnaire was made and randomly distributed on the Questionnaire Star platform. Volunteers were randomly assigned to two groups. Questionnaires were also distributed via the QQ and WeChat social apps. One hundred and fourteen volunteers were involved. The question "do videos display an AI portrait color effect?" was set to filter effective experimental questionnaires, ensuring that volunteers effectively perceived the videos displaying the product usage scenario information. Sixty-three volunteers completed the experiment. This study adopted the method of controlling variables (with and without use scenarios) to conduct the experiment, and the demographic characteristics of the volunteers involved in this experiment are shown in Table 3.

**Table 3.** Experiment samples' demographic characteristics.

| Measurement | Sample Distribution | Experiment Counts | 1 Proportion | Experiment Counts | 2 Proportion |
|---|---|---|---|---|---|
| **Gender** | Male | 20 | 31.75% | 23 | 37.70% |
| | Female | 43 | 68.25% | 38 | 62.30% |
| **Education** | Junior college and below | 13 | 20.63% | 7 | 11.48% |
| | BSc | 42 | 66.67% | 42 | 68.85% |
| | MSc | 7 | 11.11% | 11 | 18.03% |
| | PhD | 1 | 1.59% | 1 | 1.64% |
| **Disposable personal income** | Below 1000 | 9 | 14.29% | 10 | 16.39% |
| | 1001–1500 | 14 | 22.22% | 20 | 32.79% |
| | 1501–2000 | 12 | 19.05% | 12 | 19.67% |
| | 2001–2500 | 7 | 11.11% | 6 | 9.84% |
| | Above 2500 | 21 | 33.33% | 13 | 21.31% |
| **Have online shopping experiences** | cosmetic care | 50 | 79.37% | 50 | 81.97% |
| | Sport and Outdoor | 45 | 71.43% | 38 | 62.30% |
| | Underwear and accessories | 36 | 57.14% | 34 | 55.74% |
| | Jewelry | 22 | 34.92% | 11 | 18.03% |
| | Home textile products | 29 | 46.03% | 27 | 44.26% |
| | Books and Instrument | 44 | 69.84% | 44 | 72.13% |
| | Flower and Pets | 19 | 30.16% | 11 | 18.03% |
| | Digital and Home Appliance | 46 | 73.02% | 38 | 62.30% |
| | Clothes and Shoes | 53 | 84.13% | 48 | 78.69% |
| | Fresh Food | 27 | 42.86% | 31 | 50.82% |
| | Others | 23 | 36.51% | 16 | 26.23% |
| **Whether to watch short video when shopping online** | Yes | 42 | 66.67% | 42 | 68.9% |
| | No | 21 | 33.33% | 19 | 31.1% |

4.4.2. Experiment Two

The questionnaire was made and randomly distributed on the Questionnaire Star platform. Volunteers were randomly assigned to two groups. Questionnaires were also distributed via the

QQ and WeChat social apps. Seventy-four volunteers were involved. The question "do the videos display how the portable juice extractor is charged?" was set to filter out effective experimental questionnaires. Sixty-one volunteers completed the experiment. This study adopted the method of controlling variables (with and without a tutorial information) to conduct the experiment, and the demographic characteristics of the volunteers who participated in this experiment are shown in Table 3.

## 5. Data Analysis and Hypothesis Testing

### 5.1. Reliability and Validity Test

Experiments 1 and 2 were only different in terms of their video content, and the measured variables were the same. Therefore, we combined the data from these two experiments for the reliability and validity tests.

All data were analyzed by SPSS (version 25.0), which was developed by IBM in Amund City, NY, USA. The reliability coefficients of the variables in the questionnaire ranged from 0.843 to 0.954, fully satisfying the requirement of being greater than 0.7 (see Table 4), indicating that the experiments had good internal consistency.

**Table 4.** Reliability analysis of variables.

| Variables | Reliability Coefficient | Item Number |
|---|---|---|
| Perceived Information Volume (PIV) | 0.940 | 4 |
| Perceived Risks (PR) | 0.843 | 6 |
| Perceived Value (PV) | 0.954 | 10 |
| Purchase Intention (PI) | 0.902 | 4 |
| Overall Reliability | 0.926 | 24 |

All items on the measurement table were taken from similar studies with good content validity. The combined reliability of all variables was greater than 0.9, and all average variance extracted (AVE) values were greater than 0.7, indicating that the measurements have good structural validity and convergent validity (see Table 5). Furthermore, all coefficients of the variables were greater than 0.9. The square root of the AVE of all variables was greater than the correlation coefficient of this variable with other variables, indicating that this measurement has good discriminative validity (see Table 6).

**Table 5.** Combined reliability and AVE.

| Variables | Combined Reliability (C.R.) | AVE |
|---|---|---|
| Perceived Information Volume (PIV) | 0.9571 | 0.8491 |
| Perceived Risks (PR) | 0.9711 | 0.8485 |
| Perceived Value (PV) | 0.9598 | 0.7053 |
| Purchase Intention (PI) | 0.9317 | 0.7737 |

**Table 6.** The square root of AVE of all variables and coefficient matrix. The values along the diagonal are the square root of AVE.

| Variables | Perceived Information | Perceived Risks | Perceived Value | Purchase Intention |
|---|---|---|---|---|
| Perceived Information Volume (PIV) | 0.9215 | | | |
| Perceived Risks (PR) | −0.063 | 0.9211 | | |
| Perceived Value (PV) | 0.776 ** | −0.127 | 0.8398 | |
| Purchase Intention (PI) | 0.676 ** | −0.134 | 0.827 ** | 0.8796 |

** was significantly correlated at 0.01 level (bilateral).

### 5.2. Usage Scenario and Perceived Information Volume

This research explored the impacts of a video-displayed product usage scenario on consumers' perceived information volume by independent-samples *t*-tests and a variance analysis. The experimental data show that the influence of the product usage scenario information in the video ($F_{PIV}(1,61) = 23.230$, p < 0.001) significantly affected consumers' perceived information volume, as shown in Figure 8. Specifically, compared with products without usage scenario information, products with usage scenario information included in the video were associated with an increase in consumers' perceived information volume ($M_{without\ usage\ scenario} = 2.8$, $M_{with\ usage\ scenario} = 4.42$). Hypothesis 1 was verified. Displaying product usage scenario information positively affects the perceived information of consumers.

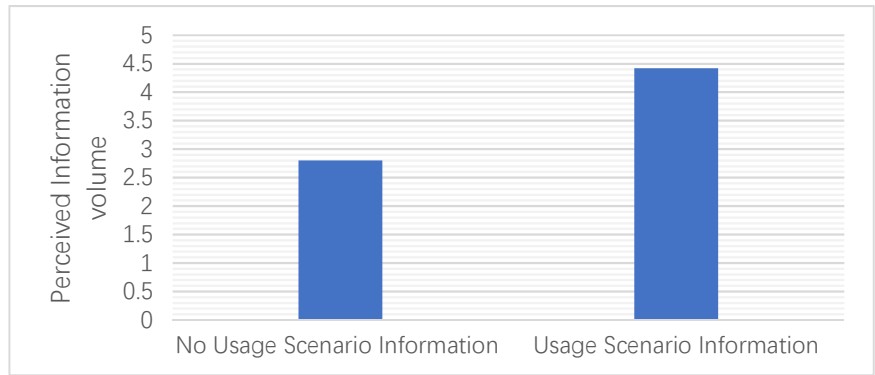

**Figure 8.** The influence of the usage scenario information on consumers' perceived information volume.

### 5.3. Tutorial Information and Perceived Information

Similar to the H1 test, this research explored the impacts of video-displayed tutorial information on purchase intention using independent-samples *t*-tests and a variance analysis. The experimental data show that the use of tutorial information ($F_{PIV}(1,59) = 0.001$, P = 0.927) in the video had no significant influence on consumers' perceived information ($M_{without\ tutorial\ information} = 4.524$, $M_{with\ tutorial\ information} = 4.522$), as shown in Figure 9. Thus, H2 failed.

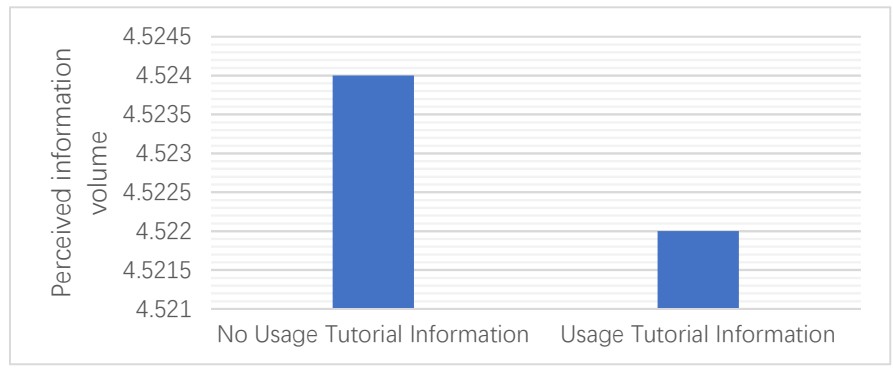

**Figure 9.** The influence of using the product's tutorial information on consumers' perceived information volume.

### 5.4. Perceived Information and Purchase Intention

A binary linear regression was used to test H3. The perceived information volume positively affected consumers' purchase intention ($F_{PIV}(1,122) = 102.672$, P < 0.001, $\beta_{PIV} = 0.676$), indicating that consumers perceptions increase when they watch short videos, increasing their purchase intension.

*5.5. Perceived Information, Perceived Risk, and Purchase Intention*

Based on the SOR model, consumers' perceived information affects their perceived risk, thereby affecting their purchase intention. Wen proposed the mediating effect test procedure [41], and a Bootstrap mediation variable test was carried out. The selected sample quantity was 5000. At a 95% confidence interval, the data showed a confidence interval for the Bootstrap test of [–0.0118, 0.0583], which includes 0, indicating that the perceived risk does not mediate the perceived information and purchase intention.

*5.6. Perceived Information, Perceived Value, and Purchase Intention*

As H5 states, consumers' perceived information influences a product's perceived value, thereby affecting purchase intention, i.e., perceived value plays a mediatory role. Similar to the test presented in Section 5.5, the selected sample quantity was 5000. At a 95% confidence interval, the data showed that perceived value mediated perceived information and purchase intention. The confidence interval of the Bootstrap test was [0.4563, 0.7598]. This excludes 0, indicating the existence of a mediating effect with a value of 0.5906. The data show that consumers' perceived information increases a product's perceived value, thereby positively affect purchasing intention.

## 6. Conclusions

The research purpose of this paper was to explore the influence of video display on consumers' purchase intentions, especially for home devices and appliances. The contributions of this research included two aspects: The first was to take the content of the video as the research object to explore its influence on consumers' perceptions and purchase intention. Second, according to the characteristics of specific product categories, we studied the factors that affect video display. Therefore, this study used the S–O–R model, from the perspective of perception, to explain how video display affects consumers' perceptions and perceived information quantity and to determine how to generate a corresponding purchase intention after a consumer receives a stimulus. Specifically, the product usage scenario information significantly affects consumers' perceived information volume. However, the tutorial information does not have the same effect. According to the results of the experiments, the consumers' perceived information volume has a positive effect on the consumers' perceived value, thereby affecting consumers' willingness to pay. However, the perceived information volume of consumers has no significant effect on consumers' perceived risk. To the end, for the digital home appliance products, video display was played to influence consumers' perceptions of information. When consumers get more product information from the video and when the consumer feels that the benefit of having the product outweighs the costs, the product's perceived value increases, and consumers will tend to want the product and have a strong purchase intention.

*6.1. Theoretical Contributions*

In terms of e-commerce video content, this paper enriches the contribution of previous studies on the influence of online product presentation formats on consumers' perception. According to the study, product usage scenario information can significantly and positively influence the perceived information quantity of consumers. This is related to the information search behavior. When consumers shop online, they search for internal and external information. Videos show more product usage scenario information, and external information stimulates consumers to conduct internal searches, thereby generating associations and increasing their perceived information volume [28]. However, the influence of tutorial information on consumer perceptions is not significant. This is probably related to consumers' knowledge and experience. Consumers are familiar with how to use devices but are concerned with other aspects, such as a portable juice extractor's portability and price.

This paper also confirmed that consumers' perceived information volume significantly affects their purchase intention. Through the experiment, this study believed that video display can effectively

influence consumers' perceptions of information. When the perceived information of a consumer increases, external search information stimulates the consumer to conduct an internal search, arousing the consumer's related imagination and encouraging them to make a purchase decision without checking out the physical products [29]. This finding is in accordance with existing studies [42]. Furthermore, this study enriches the mechanism of the influence of perceived information volume on consumers' willingness to pay. This paper explored the influence pathway through which perceived information influences consumers' purchase intention by taking the perceived value as mediating variables. In online shopping, an increase in consumers' perceived information increases their perceived value of the product, thereby positively affecting purchase intention. Consumers get a deep understanding of the product when a greater information volume is perceived. When consumers realize that it will bring more benefits when they buy this product or service, the level of their perceived value will increase [35]. This positively affect consumers' purchase intention, causing them to behave positively [37]. However, this study has a different outcome from previous studies. Many studies believed that when shopping online, consumers will perceive more risks if they cannot get enough information. This will weaken consumers' purchase intention. When consumers receive more product information, their level of perceived risk will be decreased [33]. However, although consumers perceived more volume of information when watching product presentation videos online, there was no significant effect on their perceived risk level. This may be related to perceived risk measurement variable selection. This article only explored the influences of consumers' perceived information on consumers' functional risk and social risk. It is possible that video displays for digital home appliance products have no significant influences on the functional risk and social risk.

### 6.2. Management Implications

These research results may be used by online retailers of digital appliance products during the design of their product display videos so as to improve the information perceived when consumers watch product display videos, thereby positively influencing consumers' purchase intention and, finally, achieving sales goals.

First, a video display should include information about product usage scenarios to stimulate consumers' associations with products by making it clear that the product can be effectively used in different scenarios, making consumers feel immersed, improving consumers' perceived value of the product and, finally, increasing consumers' purchase intention. In this research, the inclusion of tutorial information in a short video display of a juicer extractor did not make a significant difference to consumers' perceived information volume, possibly because consumers' existing knowledge and experience can allow them to understand how to use the product. Manual information is not widely suitable for all products from the digital appliance category. Appropriate tutorial information for short video displays should be chosen according to the characteristics of the product itself.

In addition, we emphasize the relationship between consumers' perceptions and purchase intentions. In this study, the increase in perceived information quantity and the perceived value of products shown after watching the video eventually positively affected consumers' purchase intentions and encouraged them to make positive purchase decisions. Online retailers can display information that encourages consumers to perceive the value of products in an e-commerce video by analyzing the characteristics of products and target consumers, thereby having an effective impact on consumers' purchase intentions.

### 6.3. Limitations and Further Improvements

This research has some limitations and provides an opportunity for further improvements. First of all, to avoid the interference of other factors, only videos were used in the experiments, and the simulation of real online shopping was lacking. When shopping online, consumers will not only watch videos but also search for product specifications, similar products, and other buyers' comments. Second, the selected products for the experiments had limitations. Only a smart phone and juice extractor were

selected. Thus, it is unknown whether our results can be extended to all digital household appliances. Further experiments are needed. Finally, the experimental data have some limitations. A total of 124 valid datasets were collected, which is a small amount of data. More people should be invited to participate in similar experiments in the future to ensure the reliability of experimental results.

Further researchers are encouraged to do further study on online product presentations, such as applying the fuzzy-set qualitative comparative analysis (fsQCA) method to study more complex situation. Referring to the application of fsQCA in consumer purchase, Pappas et al. combined trust, privacy, emotion, and experience to explain consumers' purchasing behavior in personalized online shopping [43]. According to this study, online product presentation videos show various elements. Maybe we can combine the products' appearances, features, brands, usage scenario information, tutorial information, production process, and host and explore how to have a more positive impact on consumers' purchase intention under different combinations. This is no longer limited to studying the relationships between variables in a single context but can also explore their impact on outcomes [44].

**Author Contributions:** Conceptualization, B.S. and R.H.; methodology, R.M.; software, R.H.; validation, R.H., R.M. and B.S.; formal analysis, R.H.; investigation, R.H.; data curation, R.H.; writing—original draft preparation, R.H. and B.S.; writing—review and editing, B.S. and R.H.; funding acquisition, R.M. and B.S.

**Funding:** This research was funded by the Chinese National Funding of Social Sciences (No.14AGL023) and the Fundamental Research Funds for the Central Universities (No.2019 CDJSK 02 PT 19).

**Acknowledgments:** Thanks to all commenters for their valuable and constructive comments.

**Conflicts of Interest:** The authors declare no conflict of interest.

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
