# Peer review of "Impacts of Video Display on Purchase Intention for Digital and Home Appliance Products—Empirical Study from China"

_futureinternet, doi:10.3390/fi11110224_

Round 1

Reviewer 1 Report

The paper seeks to examine how product videos can influence consumers purchase intention when shopping for digital and home appliances. The topic is very interesting and the authors run 2 experiments to collect data for 2 different types of products in China. The paper suffers from a some major flaws. 

First of all, the motivation and research gap are not properly addressed in the paper. Once the topic and problem have been introduced you need to discuss what other studies have done/found about this specific issue. This is now missing from the paper. Note that the authors go to the literature only to justify the importance of using videos to get more information about a product. Here you need to introduce the main factors that you will study and explain what has been done in that area. That would lead to the gap. Then explain why this is important to address and describe how you plan to do it here. That will lead to your contribution and novelty. Using numbers from recent reports as you do now, also helps but it is not enough.

In section 2 you need to provide more background and related work around the concepts you examine. Here it would be appropriate to introduce the SOR model that you use in the study and provide the necessary background.

Table 1 and 2 can be merged to avoid repetition. 

Validity test results are given only for one of the experiment (table 6). Which one? Why not for both?

The paper needs to add a discussion of the findings and try to connect with the literature in their theoretical contributions.

Take a look at this paper (outline, presentation, and style of writing) as I think it can help.

Pappas, I. O., Kourouthanassis, P. E., Giannakos, M. N., & Chrissikopoulos, V. (2017). Sense and sensibility in personalized e‐commerce: How emotions rebalance the purchase intentions of persuaded customers. Psychology & Marketing, 34(10), 972-986.

The paper has language issues that make it difficult to read.

Reviewer 2 Report

The paper provides an original and interesting analysis on the content of videos.

I find the introduction not informative enough in providing the managerial and theoretical implications.

Another important element to enrich is the literature review. First, I suggest to create a literature review table that will offer readers a clear indication of what had been done and a positioning for your paper. Second, I would amend what you call “Modelling”. It is actually a theoretical framework. Finally, I suggest to integrate the literature with some papers on video storytelling (which show the power of videos compared to other online sources such as photos or text). Few examples here are: Exploring how video digital storytelling builds relationship experiences in P&M and The determinants of stakeholder engagement in digital platforms in Journal of Business Research. 

Overall, I find the experimental part convincing. It is although important that you explain quickly how you controlled the different elements that vary in videos (compared to photos and videos this is much harder to achieve).

Finally, the end part of the paper can be enriched. If you do the literature review table as suggested, this part can be enriched by discussing the previous tension in the literature.

Round 2

Reviewer 1 Report

I would like to thank the authors for working on their revised manuscript. One of the main concerns that I had in the previous round was the weak introduction of this paper. In the updated version only a couple of lines have been added simply mentioning 4 references. Please look at my comments in the previous round and try to expand a bit your introduction.

The background has been improved. 

Some theoretical implications have been added now as they have been significantly extended. Try to connect the findings more with the literature that is reviewing in the previous sections.

Future work can be extended as now it is very limited. The authors and this study would benefit by performing a fuzzy set qualitative analysis (fsQCA). Looking at the findings running an fsQCA analysis can lead to more and new insight (and a new paper). Here, this should be mentioned in the future work (which now is quite limited). 

See the following papers that employ the method and based on them you describe in detail why and how fsQCA can be employed in your context.

Pappas, I. O., Kourouthanassis, P. E., Giannakos, M. N., & Chrissikopoulos, V. (2016). Explaining online shopping behavior with fsQCA: The role of cognitive and affective perceptions. Journal of Business Research, 69(2), 794-803.

Pappas, I. O. (2018). User experience in personalized online shopping: a fuzzy-set analysis. European Journal of Marketing, 52(7/8), 1679-1703.

Woodside, A. G. (2014). Embrace• perform• model: Complexity theory, contrarian case analysis, and multiple realities. Journal of Business Research, 67(12), 2495-2503.

Reviewer 2 Report

I am happy with the reviews and suggest acceptance at this stage

Author Response

We are very excited about your acceptance of our manuscript. Your opinions have greatly improved our research level. Finally, thank you again for giving us the opportunity to publish our paper.

Round 3

Reviewer 1 Report

I would like to thank the authors for addressing my comments.